# The Tumor Microenvironment of Medulloblastoma: An Intricate Multicellular Network with Therapeutic Potential

**DOI:** 10.3390/cancers14205009

**Published:** 2022-10-13

**Authors:** Niek F. H. N. van Bree, Margareta Wilhelm

**Affiliations:** Department of Microbiology, Tumor and Cell Biology (MTC), Karolinska Institute, 17165 Stockholm, Sweden

**Keywords:** medulloblastoma, tumor microenvironment, age-associated differences, extracellular matrix, immune cells, brain tumor vasculature, leptomeningeal dissemination

## Abstract

**Simple Summary:**

The current treatment options for medulloblastoma, the most common malignant childhood brain cancer, are associated with many negative side effects and toxicities. Therefore, novel treatment options are needed that target the tumor without affecting the healthy tissue. Medulloblastoma tumors consist of a wide variety of cell types and extracellular components that make up the microenvironment of the tumor. This tumor microenvironment influences the development, progression, and relapse of medulloblastoma through different cell–cell and cell–extracellular matrix interactions. Obtaining insights into these interactions will help with gaining a better understanding of this malignancy. Additionally, it could support the search for new targets of treatments directed at components of the tumor microenvironment.

**Abstract:**

Medulloblastoma (MB) is a heterogeneous disease in which survival is highly affected by the underlying subgroup-specific characteristics. Although the current treatment modalities have increased the overall survival rates of MB up to 70–80%, MB remains a major cause of cancer-related mortality among children. This indicates that novel therapeutic approaches against MB are needed. New promising treatment options comprise the targeting of cells and components of the tumor microenvironment (TME). The TME of MB consists of an intricate multicellular network of tumor cells, progenitor cells, astrocytes, neurons, supporting stromal cells, microglia, immune cells, extracellular matrix components, and vasculature systems. In this review, we will discuss all the different components of the MB TME and their role in MB initiation, progression, metastasis, and relapse. Additionally, we briefly introduce the effect that age plays on the TME of brain malignancies and discuss the MB subgroup-specific differences in TME components and how all of these variations could affect the progression of MB. Finally, we highlight the TME-directed treatments, in which we will focus on therapies that are being evaluated in clinical trials.

## 1. Introduction

Medulloblastoma (MB) develops in the cerebellum and is the most common form of malignant brain cancer in children. While MB predominantly arises in infants and children, rare cases of adult MB also occur [1]. This diverse group of tumors can be divided into four major subgroups based on molecular and clinical characteristics, namely wingless (WNT), sonic hedgehog (SHH), group 3, and group 4 [2]. Additionally, these four subgroups can be further subdivided into twelve subtypes [3]. The WNT-MB and SHH-MB subgroups are characterized by deregulation of the WNT signaling and SHH signaling pathways, respectively. The underlying molecular mechanisms for group 3 MB and group 4 MB are not entirely understood. However, recent studies have indicated that group 3 MB contains photoreceptor subpopulations, while group 4 MB mainly expresses a neuronal glutamatergic gene signature [4,5]. It is thought that every MB subgroup arises from a specific location in the cerebellum with its own neural progenitor cell type of origin [6,7]. SHH-subgroup MB, for example, typically arise in the lateral cerebellar hemispheres, where granule cell precursors are postulated as the cell of origin. Evidence was provided by mouse model studies [8,9] as well as by comparative single-cell transcriptome sequencing (scRNA-seq) of developing postnatal granule cerebellar and MB cells [10]. It was also demonstrated that the granule cell precursors in MB show less differentiation compared to healthy granule cell precursors, presenting more proof for the tumorigenic origin of SHH-MB. Lower rhombic lip precursors have been appointed as the cells of origin for WNT-subgroup MB in both mouse and human MB tumor samples [6,11]. Precursors for both group 3 and group 4 MB are thought to arise from excitatory cerebellar neurons (eCN) and unipolar brush cell (UBC) precursors, which was demonstrated in a recent study where the transcriptomic landscape of primary MB tumors was compared with a human fetal cerebellum scRNA-seq reference set to avoid cross-species comparisons [11]. This study has further revealed that group 3 MB cells are more confined to the early progenitor-like eCN/UBC trajectory, beginning with rhombic lip precursors, which then transitions midway to more differentiated eCN/UBCs that are associated with group 4 MB, demonstrating that group 3 and group 4 MB tumors exist along a transcriptional continuum that reflects early human cerebellar development. This was confirmed in two studies comparing human scRNA-seq MB profiles with developing murine cerebellar scRNA-seq reference datasets [7,12]. Group 3 MB displayed most resemblance with Nestin^+^ stem cells and group 4 MB was highly similar to UBCs. The difference in cell of origin between MB subgroups define this heterogenous disease’s unique molecular and clinical features. 

MB is fatal if left untreated. The current treatment for MB includes multimodality therapy consisting of maximal surgical removal of the tumor followed by high-dose cytotoxic chemotherapy and craniospinal radiation treatments for patients older than three years. These treatment modalities have increased the overall survival rates for MB by up to 70–80% [13]. However, the standard-of-care treatments may also lead to devastating side effects, mainly in young survivors, due to impairment of the developing pediatric brain. The quality of life can be severely impacted by cognitive deficits, endocrine disorders, and an increased incidence of secondary tumors later in life [14]. Moreover, patients classified as high-risk, such as those with metastatic spread or incomplete tumor resection, continue to have a poor prognosis [15]. The recurrence of disease in MB patients is almost always fatal and occurs in 30% of patients [16]. Factors that influence the time to relapse are the molecular MB characteristics and if the patient received upfront craniospinal irradiation. All of this indicates that novel therapeutic approaches for MB are needed. 

Besides the direct targeting of tumor cells, strategies to target components of the multicellular environment around the tumor have emerged as promising therapeutic approaches to cancer treatment [17]. The tumor microenvironment (TME) plays a key role in regulating tumor formation, progression, and spread and influences therapy responses to standard-of-care treatments. The TME consists of immune cells, including tumor-associated macrophages (TAM), T and B lymphocytes, dendritic cells, neutrophils, natural killer (NK) cells; supporting stromal cells, such as cancer-associated fibroblasts (CAF), mesenchymal stromal cells, and pericytes; blood and lymphatic vessels; the extracellular matrix (ECM); and signaling molecules such as cytokines, chemokines, and growth factors. Even more complex is the TME of brain tumors, because in addition to this vast display of components the brain TME also contains unique brain-resident cell types such as microglia, astrocytes, and neurons and distinctive vasculature systems such as the blood–brain barrier (BBB) [18]. In this review, the components of the TME of MB will be discussed (Figure 1). Since MB is mainly a pediatric tumor, emphasis will be placed on the effect age has on the TME. Furthermore, TME-directed therapies will be reviewed with a focus on treatments that are being assessed in clinical trials.

## 2. The Impact of Age on the Tumor Microenvironment of Brain Malignancies

Cancer is mainly described as an aging-associated disease. The incidence of most cancers increases dramatically as we age due to the accumulation of mutations over time [26], demonstrating the critical role age plays in tumor development. However, malignant tumors such as MB mainly occur in children, which indicates that age is not the only driving force for tumor development. It has been shown that the age of the TME can influence tumor progression and that there are differences in young and adult TME regarding the ECM composition, presence of senescent cells, immune microenvironment, and growth factors [27]. The majority of studies on the TME have been conducted in adult carcinomas, such as breast cancer. Therefore, the role of each stromal component in these adult cancers is better understood [28]. The characterization of the microenvironmental components in pediatric cancers such as MB and a better understanding of the differences between young and adult TME are key to increasing our understanding of molecular mechanisms regulating MB development and identifying novel therapeutic targets. 

Cellular senescence is a process where cells go into irreversible proliferative arrest due to specific stressors such as DNA damage without undergoing apoptosis. This process is mainly associated with an aged TME [29]. Senescent cells undergo morphological and metabolic changes and can promote tumorigenesis mainly by activating a senescence-associated secretory phenotype (SASP). This SASP secretome consists of growth factors, chemokines, proinflammatory cytokines, and proteases and is strongly dependent on the location where the senescent cell arises [30]. The SASP promotes tumor growth, invasion, and metastasis by inducing angiogenesis, promoting immune evasion and paracrine signaling towards viable neoplastic cells and other cells in the TME [31]. Although senescent cells are mainly observed in the adult TME, studies have also proven the involvement of senescence in pediatric tumors. Buhl et al. demonstrated that the SASP mediates oncogene-induced senescence (OIS) in pediatric pilocytic astrocytoma, a low-grade glioma [32]. These malignancies grow slowly, which is thought to be due to OIS. Interestingly, the high expression of a SASP gene expression profile, in particular the SASP component *IL1B*, was associated with a high progression-free survival rate, independently of the status of tumor resection, showing that the SASP can also have anti-tumorigenic effects. Pilocytic astrocytoma is a relatively treatable form of pediatric brain cancer. However, this finding could also explain why many other brain malignancies stay dormant for a long time. The development of senescence in malignant cells, avoiding growth and detection, could explain why some MBs are only detected at later stages in life, even though MB is an embryonal tumor. The malignancy of these tumors can ultimately increase by evading OIS. Evidence for this mechanism of progression was given by Tamayo-Orrego et al., who demonstrated that spontaneous mutations of *TP53*, a mutation mainly found in childhood MB compared to adult MB, cause the evasion of senescence, leading to MB progression [33]. MB tumors with this mutation had lost the expression of the senescence markers p16^INK4a^ and p21^Cip1^, whereas preneoplastic MB lesions displayed elevated levels of both markers. Furthermore, Pallavicini et al. found that senescence and apoptosis via p53 could be induced by the inactivation of citron kinase (CITK) [34]. They demonstrated that CITK deletion in NeuroD-SmoA1 transgenic mice led to decreased MB tumor growth and increased overall survival, which were associated with an increase in apoptotic cells and the expression of senescence markers p16^INK4a^, p21^Cip1^, and p27^Kip1^ (Figure 2). In addition to OIS, the survivors of pediatric cancers also show signs of therapy-induced senescence, which results in a higher risk of health complications later in life [35,36].

Another major difference between the young and adult TMEs is the composition of the ECM. This matrix is secreted by cells and is responsible for the structural and biochemical integrity of most tissues in the human body, with a large diversity in composition across organs. The ECM is constantly changing due to deposition and degradation according to a highly dynamic process. A decreased ECM integrity is strongly associated with aging, but also with tumor progression and metastases [37,38]. In addition, it has been shown that senescent cells play a role in matrix stiffening and ECM remodeling in the local TME of age-related diseases [39,40]. For instance, senescent fibroblasts promote branching morphogenesis in primary breast cancer organoids through their elevated secretion of matrix metalloproteinase-3 (MMP-3) [40]. MMP-3 and related enzymes mediate the increased degradation of multiple ECM components such as collagens, laminin, and proteoglycans, causing matrix stiffening. Interestingly, it was found that the tissue inhibitors of metalloproteinases (TIMPs) TIMP-2 and TIMP-3 are frequently methylated in MB, and their expression decreases in adult SHH-MB compared with infant SHH-MB (Figure 2) [41,42,43]. The ability to modify the stiffness and crosslinking of the ECM allows the spread and progression of tumor cells.

The immune microenvironment is also strongly influenced by the age of the TME. Persistent low-grade inflammatory responses increase with aging [45]. This process of chronic inflammation, called inflammaging, is strongly related to cancer progression due to the disturbance of the acute inflammation and its effects on tissue deterioration [46]. Cellular senescence plays a key role in inflammaging through SASP induction, which is responsible for a continual increase in inflammatory cytokines, such as IL-6, interferon-gamma, and tumor necrosis factor. A second process that intensifies during aging is called immuno-senescence, an age-related disturbance of the immune system exhibiting a decline in overall immune function and immune cell populations [47]. Interestingly, pediatric tumors such as MB are considered ‘cold’ tumors, with already limited infiltration of cytotoxic lymphocytes in the TME. MB especially displays low levels of tumor-infiltrating lymphocytes (TILs), as demonstrated by a pediatric pan-central nervous system tumor analysis of immune cell infiltration [23]. However, significant differences were present in the MB subgroups, with the infant SHH-MB subgroup displaying the highest levels of immune cell infiltration (Figure 2). 

### Insights in TME Age-Related Differences in MB with Transcriptomics

Multiple sequencing methods have given more insight into the differences between pediatric and adult tumors and their TMEs. For instance, scRNA-seq has become known as a powerful tool for studying the cellular state and obtaining a better understanding of cells in the context of their microenvironment. Recently, scRNA-seq studies of primary MBs revealed subgroup-specific single-cell heterogeneity in these tumors [5,7,12,48]. Cerebellar granule cell precursors have been confirmed as the cells of origin for SHH-MB in all these studies. The SHH subgroup has the biggest age variation of all MB subgroups, displaying a bimodal distribution, comprising the majority of infant (≤3 years) and adult (≥17 years) MBs. Interestingly, age-associated categories could be made of the SHH-MBs based on the differentiation state of the granule cell precursors [12]. Adult SHH-MB tumors correlated with a higher fraction of undifferentiated granule cell precursors (marked by high expression of *ATOH1*), whereas granule neuron cell precursors of pediatric SHH-MB tumors displayed more of an intermediate to mature differentiation state (marked by high expression of *NEUROD1*). The differences in differentiation states could be caused by age-related alterations in the TME, since granule cell precursor differentiation is strongly dependent on secreted factors of the TME.

Comparisons of the molecular signatures of pediatric and adult SHH-MB by DNA methylation and transcriptional profiling studies have also revealed age-associated variations [42,49]. A gene set enrichment analysis showed that genes related to the composition and functions of the ECM are highly upregulated in pediatric SHH-MBs compared to adult SHH-MBs (Figure 2) [42]. These genes encode collagens (*COL1A1*, *COL3A1*, *COL4A1*), laminins (*LAMA1*, *LAMA4*, *LAMB1*), cell adhesion molecules (*CADM2*, *CDH11*, *PECAM1*), and other ECM structural components such as glycoproteins (*SMOC2*, *SPARC*, *FN1*) and proteoglycans (*LUM*, *FREM2*). Adult SHH-MBs have a higher overall mutational load than childhood SHH-MBs, which corresponds to the higher mutational burden seen in other adult tumor malignancies [50,51,52]. Interestingly, specific SHH-pathway-associated mutations could be assigned to different age groups. *PTCH1* mutations occurred at similar frequencies in pediatric and adult SHH-MB [53]. However, infants (younger than three years) showed a higher incidence of mutations in *SUFU*, while *SMO* mutations were predominantly found in adults. Mutations in both *SUFU* and *SMO* were less frequently found in children between 4 and 17 years old. The TME plays an important role in regulating tumor progression through these mutationally activated signaling pathways. This has been shown in other malignancies, such as prostate and pancreatic cancers, where SHH produced by tumor cells communicates with the stromal cells, such as mesenchymal and endothelial cells, promoting tumor growth, metastasis, lymphangiogenesis, and perineural invasion due to paracrine signaling [54,55,56,57]. Other aberrations enriched in childhood and adult SHH-MB are mutations in *TP53* and the *TERT* promoter, respectively (Figure 2) [51,53]. Both mutations have been hypothesized to have roles in escaping senescence in MB [44]. These studies underline the importance of signaling pathways in the crosstalk between cancer cells and the TME.

## 3. The Extracellular Matrix Composition of Medulloblastoma

In addition to the unique brain-resident cell types that are present in the brain TME and the physical protection by the BBB, the composition of the ECM in the brain is also considerably different from other organs. The normal ECM environment of the brain is mainly composed of glycoproteins, proteoglycans (mainly chondroitin sulfate and heparan sulfate proteoglycans), glycosaminoglycans, and growth factors, and contains only low levels of fibrous proteins (e.g., fibronectin and collagen), in contrast to the ECM compositions in many other organs [58]. However, the brain ECM changes its composition drastically during tumor development through aberrant ECM deposition, post-translational modifications, proteolytic degradation, and force-mediated physical remodeling, resulting in a loss of ECM integrity. The mechanisms of ECM remodeling in tumor progression and metastasis have been thoroughly described in a review by Winkler et al. [59]. The ECM network is involved in brain development and tissue integrity by providing biochemical and structural support for cells. The loss of this tissue ECM integrity is associated with an accumulation of ECM components and is strongly linked with tumor progression [60]. Therefore, dysregulation in the ECM’s composition is considered one of the hallmarks of developing a premetastatic niche [61]. A mechanism of loss of ECM integrity in MB was described by Ridgway et al. [24]. They demonstrated that extracellular heparanase, the major enzyme responsible for the degradation of heparan sulfate, regulates intracellular SHH and WNT3A signaling in human MB cells by altering the localization and expression of the GLI transcription factors and β-catenin. Furthermore, it has been shown that pediatric brain tumors contain higher levels of heparanase compared to healthy brain tissue [62]. Treatment with the heparanase inhibitor PG545 resulted in the selective killing of pediatric brain tumor cells and reduced migratory abilities and in vivo tumor growth. 

Both in healthy tissue and tumors, the major producers of ECM are fibroblasts. However, fibroblasts are nearly absent in the central nervous system [63], which explains why the existence of CAFs in MB has not been studied. Nonetheless, some studies claim to have identified CAFs in glioblastoma [64,65,66], indicating that the presence of CAFs in MB might not be completely overlooked. 

Because of its distinctive ECM composition, the assessment of the stromal compartments of brain tumors provides the potential for diagnostics and the detection of novel targets. However, only limited studies have been performed by analyzing the ECM compositions in brain tumors. The first comparative analysis of the ECM proteomes of two brain malignancies was recently performed by Trombetta-Lima and colleagues [67]. In their study, the proteome profiles of MB and glioblastoma were compared to the cerebellum and neocortex, respectively. Both tumor types displayed distinct ECM signatures when compared to their respective controls and between malignancies. The MBs presented with an ECM profile enriched in fibrous proteins such as collagens (among others COL1A1, COL5A2, COL6A3), glycoproteins (i.e., fibrillins), and proteoglycans (lumican). Genes encoding for these proteins were upregulated, particularly in SHH and WNT-MB subtypes. It is interesting to know that collagens can have immune-modulatory functions within the TME, which could contribute to the immunosuppressive environment seen in MB. Immune-modulatory properties of collagen have been described in several types of cancers, such as lung, colorectal, ovarian, and pancreatic cancers, in which high-density collagen drives macrophage polarization towards a TAM immunosuppressive phenotype [68,69,70] and suppresses T-cell migration, infiltration [71,72,73], and activity [74,75]. The mechanisms of how collagens modulate immune properties in cancer are elaborately described in a review by Rømer et al. [76]. 

In addition to the specific MB ECM profile, it has been shown that the glycoproteins laminin and vitronectin can be used to distinguish MB subgroups from each other [77]. Vitronectin was highly expressed in group 3 MB tumors relative to SHH and WNT-MB tumors and showed only intermediate expression in group 4 MB tumors. In contrast, laminin (isoforms 111 and 211) was highly expressed in SHH-MB compared to the other subgroups. Furthermore, the gene expression levels of *VTN* and *LAMA1/LAMA2* helped in predicting patient survival outcomes, where vitronectin-expressing group 3 and laminin-expressing group 4 MB patients were designated as high-risk groups with poor survival [77]. This indicates that the tumor ECM profiles of MB can be used as a prognostic factor and as potential novel therapeutic targets for MB. However, more research is needed to find effective targets for stromal-targeting therapies in MB.

## 4. The Immune Cell Landscape of Medulloblastoma

A diverse immune microenvironment is present in brain tumors, which interacts with malignant cells through an intricate network that can promote and inhibit tumor progression. Immune cells such as TAMs, microglia (specialized macrophage-like cells in the central nervous system), T and B lymphocytes, NK cells, dendritic cells, and neutrophils contribute to tumorigenesis in unique ways [78]. Tumors can be described as either ‘hot’ or ‘cold’ tumors based on the probability of triggering a strong immune response. Hot tumors usually respond better to immunotherapy due to the accumulation of cytotoxic lymphocytes and proinflammatory cytokines in the tumor [79]. In contrast, cold tumors lack these characteristics and tend to have a more immune-suppressive tumor environment. As mentioned earlier, compared to other types of solid tumors, MB is a cold tumor with a low occurrence of infiltrating immune cells [5,21,23,80,81]. Furthermore, there is considerable heterogeneity in infiltrating immune cells between MB subgroups. The characterization of infiltrating immune cells in immunocompetent SHH and group 3 MB animal models revealed that murine SHH-MB tumors contained more dendritic cells, myeloid-derived suppressor cells, TAMs, and TILs, whereas group 3 MB tumors were composed of more CD8^+^ T-cells [80]. An analysis of the subgroup-specific immune microenvironment in human MB tumors was based on gene expression profiles [21], spatial protein expression, and cytokine secretion profiling [81], and confirmed the data from the animal models. Human SHH-driven MB tumors recruited more TAMs and T-cells, whereas group 3 and group 4 MBs contained more CD8^+^ T-cells and cytotoxic lymphocytes. Moreover, it was shown that group 4 MB tumors had significantly larger populations of neutrophils and CD4^+^ T-cells compared to the other subgroups. WNT-driven MB tumors were not enriched for any type of immune cell compared to the other subgroups. However, it must be noted that the WNT subgroup was either not included or excluded due to the limited sample size in several of these studies [5,80,81], making it difficult to formulate a clear statement about the immune cell microenvironment of WNT-driven MB.

The most abundant immune cells in MB tumors are tissue-resident microglia and TAMs [21]. Therefore, the research has mainly focused on deciphering the role of these immune cells in the TME of MB. TAMs can stimulate tumor growth by suppressing T-cell activity, promoting angiogenesis, and creating an immunosuppressive TME through the production of growth factors, cytokines, and chemokines [82]. However, a subset of TAMs can also achieve anti-tumoral effects through the production of pro-inflammatory cytokines. It has been reported that SHH-MB tumors have greater infiltration of TAMs, as well as greater expression of TAM-associated genes (CD163 and CSF1R), compared to other MB subgroups [83]. Furthermore, distinct subsets of TAMs derived from circulating monocytes and microglia were identified within the SHH subgroup [84]. Upon radiation treatment, immunosuppressive TAMs were recruited that reduced neutrophil and T-cell infiltration in MB. However, in both these studies, the role of TAMs on SHH-MB initiation and progression was not further elucidated. 

Whether TAMs support or inhibit the growth of MB remains controversial. Maximov et al. demonstrated that TAMs could inhibit SHH-MB tumor growth by promoting tumor cell death both ex vivo and in vivo, unlike their pro-tumoral role in glioblastoma [85]. In SHH-MB, activated microglia are responsible for the recruitment of bone marrow-derived macrophages to the TME via the production of CCL2. Furthermore, they showed that the survival of B6 WT pups injected with *NeuroD2:SmoA1*-derived tumors significantly decreased after treatment with two CSF1R (a survival factor for both monocyte-derived macrophages and tissue-resident microglia) inhibitors. Treatment with CSF1R inhibitors could also not prevent the recurrence and metastatic spread of MB [86]. In contrast, a pro-tumoral role of TAM infiltration in SHH-MB was explained by others [87,88]. Tan et al. have used an *Atoh1-SmoM2* immunocompetent mouse model that develops sporadic SHH-MB to reveal these pro-tumoral effects [87]. In addition, they have shown that treatment of the tumor-bearing *Atoh1-SmoM2* mice with a CSF1R inhibitor prolonged survival. These studies clarify three points: (1) it is crucial to consider the immune profile of the model when a model is being selected to research tumor–TME interactions; (2) suitable syngeneic models are needed to faithfully recapitulate MB development; (3) more research is needed to fully understand the function of TAMs in MB.

### Immunotherapy against Medulloblastoma

The minimal mutational tumor burden and low levels of infiltrating immune cells in MB challenge the discovery of targets suitable for immunotherapy in this malignancy [89]. The comprehensive review by Hwang and colleagues describing the current immunotherapy landscape for pediatric brain tumors gives more information about the different types of immunotherapies and related obstacles in using this type of treatment for brain cancers such as MB [90]. Clinical trials with immunotherapy so far have not uncovered compelling results in MB. Treatment with immune checkpoint inhibitors such as avelumab (PD-L1 inhibitor; NCT03451825, [91,92]), nivolumab with or without ipilimumab (PD-1 and CTLA-4 inhibitors, respectively; NCT03130959, [93]), and indoximod (IDO inhibitor, NCT02502708, [94]) were in general well-tolerated, but no significant increase in overall survival of MB patients was observed. However, it must be mentioned that in these studies the baseline T-cell infiltration and heterogeneity in the tumors were not analyzed or reported. This might have introduced bias by selecting patients who are not responsive to immune checkpoint inhibition. Due to the favorable tolerance, continuous improvements in these types of treatments, and a more thorough selection of patients, immune checkpoint inhibitor treatments are still worth pursuing further. Therefore, many phase 1 and phase 2 clinical trials are currently ongoing that include MB patients to further assess the potential of immunotherapy for the treatment of this malignancy (Table 1). The majority of these studies evaluate the efficacy of immune checkpoint inhibitors, such as anti-PD-1 and anti-PD-L1, or chimeric antigen receptor (CAR) T-cell therapies that target specific tumor-associated antigens. CAR targets that are commonly found among pediatric brain tumors are B7-H3, GD2, IL-13Rα2, EphA2, and HER2 [95]. An analysis of a panel of 49 pediatric brain tumor patient-derived orthotopic xenografts (PDOX), including 24 MBs, indicated heterogeneous antigen cell surface expression among the PDOXs, which is representative of the heterogeneity among pediatric brain cancer patients. PDOX samples were considered antigen-positive when ≥10% of the analyzed tumor cells were positive. B7-H3 (95.8%) and GD2 (87.5%) were most often expressed within the 24 examined MB PDOX samples, followed by IL-13Rα2 (75.0%), EphA2 (33.3%), and HER2 (16.7%) [95]. B7-H3, GD2, IL-13Rα2, and HER2 are some of the tumor-associated antigens that are currently being tested in clinical trials against MB for their potency and safety as CAR targets (Table 1). Other CAR targets that are tested in clinical trials are EGFR, NKG2L, PRAME, and cancer testis antigens (CTAs), such as WT1 and BIRC5 (Table 1; NCT03652545). Moreover, the safety of vaccine therapies and oncolytic virus therapies against pediatric brain tumors is being evaluated. The preliminary results of a phase 1 trial (Table 1; NCT03299309), in which the safety and feasibility of a novel peptide vaccine (PEP-CMV) directed against cytomegalovirus (CMV) antigen pp65 were assessed, demonstrated that PEP-CMV is well-tolerated in children and young adults with recurrent malignant glioma and MB [96]. A multi-institutional phase 2 clinical trial (Table 1; NCT05096481) was opened because of these positive results, which will examine whether PEP-CMV can serve as a novel immunotherapeutic approach for pediatric patients with high-grade glioma, diffuse intrinsic pontine glioma, or recurrent MB. Oncolytic viruses are naturally existing or genetically engineered viruses that can selectively infect and kill cancer cells without harming healthy cells. Preclinical studies showed that oncolytic viruses such as reoviruses [97], adenoviruses [98], measles viruses [99], and herpes simplex viruses (HSVs) [100] can be utilized to treat MB, which has driven the initiation of clinical trials that investigate oncolytic viruses as therapeutic options for MB (Table 1). The recent results from a phase 1 clinical trial (NCT02444546) demonstrated that treatment with engineered wild-type reovirus and sargramostim (GM-CSF) was well-tolerated in patients with recurrent or refractory disease [101]. However, this treatment could not prevent the progression of the disease, to which all patients succumbed at a median of 108 days after recruitment. Further clinical investigation is justified because of the high tolerance for this treatment. It will be necessary to determine a maximal tolerated dose and study the therapeutic modality in earlier stages of disease, since the heavy pretreatment of the patients could have affected the efficacy of this oncolytic viral approach.

A better understanding of the brain-specific immune profiles, immunosuppressive microenvironment, candidate targets, and potential combination approaches is needed for immunotherapy to be a valuable addition to the current treatment modalities of MB. Patient stratification is one of the critical steps used to assess the safety and efficacy of immunotherapy-based approaches, to make sure that the enrolled patients will actually benefit from the therapy. For this reason, clinical trials using CAR T-cells targeting HER2, EGFR, GD2, IL-13Rα2, and NKG2DL (NCT03500991, NCT03638167, NCT04099797, NCT04510051, NCT05131763, respectively) require histological evidence as one of the inclusion criteria. Furthermore, many of the clinical trials testing vaccine, oncolytic virus, and immune checkpoint inhibitor treatments (Table 1) require tissue sampling and a subsequent histological analysis before enrollment to provide the highest achievable outcome for patients. Here, it will be critical to identify appropriate predictive biomarkers for the development of personalized treatment plans. All currently ongoing clinical studies will assist in elucidating the efficacy of the varied immunotherapy strategies for MB patients (see Table 1 for a comprehensive list of ongoing clinical trials with MB patients).

## 5. Involvement of Non-Hematopoietic CNS Cells in Medulloblastoma

### 5.1. Astrocytes in MB Tumor Progression and Relapse

Non-hematopoietic cells such as astrocytes and neurons are also involved in the TME of MB. One of the most abundant cell types within the cerebellum are astrocytes, comprising Bergmann glia, granular layer astrocytes, and fibrous astrocytes [102]. Recently, they have been associated with crucial functions in the regulation of MB tumor growth, more specifically in the progression of the SHH-MB subgroup. Complement C3a, a protein enriched in human MB tumor samples, was found to be able to activate astrocytes via the p38 MAPK pathway [103]. Subsequently, these C3a-activated astrocytes promote MB tumor progression both in vitro and in vivo through TNF-α secretion. Activated tumor-associated astrocytes sustain the proliferation of MB tumor cells also through the secretion of SHH ligands, cytokines, and other ECM components [22,104,105]. SHH ligands secreted by tumor-associated astrocytes promote Nestin expression in SHH-MB tumor cells, an enhancer of the hedgehog signaling pathway and MB tumor growth [104]. This mechanism is dependent on smoothened activation, but independent of Gli1. Furthermore, the secretion of TME-promoting ECM components such as fibronectin and collagens by tumor-associated astrocytes is positively regulated by astrocyte-secreted SHH [105]. Moreover, it has been shown that tumor-associated astrocytes can produce CCL2, which helps to maintain the stem-like properties of metastatic MB cells [22], in addition to being a chemoattractant for TAMs and microglia [106,107]. The maintenance of metastatic MB cell stemness by tumor-associated astrocyte-secreted CCL2 is controlled via the JAK2/STAT3-mediated activation of Notch signaling [22]. Tumor cells can subsequently produce growth factors and cytokines, such as IL-6, which activate and recruit astrocytes. Astrocytes are also activated by tumor cell-derived TGLI1, MIF, IL-8, IL-1β, and TNF-α, as seen in lung and breast cancer brain metastases [108,109,110]. This results in a proliferative loop, increasing the malignancy of MB.

Besides their function in MB progression, tumor-associated astrocytes have also been associated with MB relapse. The clusters of quiescent SOX2+ cells have been discovered to be driving relapse in SHH-MB. SOX2+ cells can produce fast-dividing neuron (DCX- or NeuN-expressing) and glial (GFAP- or S100-β-expressing) progenitor-like cells [111]. Additionally, it was found that SOX2+ cells with an astrocyte-like transcriptome contribute to the relapse of MB through the non-canonical activation of GLI signaling [112]. This non-canonical GLI activation downstream of SMO was found to be dependent on MYC. Treatments with inhibitors that target the upstream SHH signaling components, such as the SMO inhibitor vismodegib, cause an enrichment of tumor-associated astrocytic SOX2+ cells resulting in an increased chance of relapse. Interestingly, it has been shown that the *MYC*-driven transformation of SOX2+ astrocyte progenitor cells give rise to group 3 MB [113], suggesting that SOX2+-tumor-associated astrocytes might have different tumorigenic functions in the MB subgroups.

Lastly, it has been demonstrated that the trans-differentiation of a fraction of SHH-MB cells into tumor-associated astrocytes is a novel mechanism through which MB tumor progression and relapse are promoted [25,114]. Trans-differentiation is a process in which one specialized cell type changes into another without entering a pluripotent state [115]. Mechanistically, the phosphorylation of SOX9 in MB cells is stimulated by bone morphogenetic proteins (BMPs), which is required for trans-differentiation into tumor-associated astrocytes [114]. These trans-differentiated astrocytes secrete IL-4 and polarize tumor-associated microglia to produce IGF-1, which in turn stimulates tumor progression by accelerating migration and adhesion [25,116]. All of these mechanisms highlight additional ways how the brain TME is modeled by MB cells and should be investigated to find putative targets for treatment. Moreover, more studies are needed to examine the role of astrocytes in WNT, group 3, and group 4 MB.

### 5.2. Tumorigenic Activity of Neurons

The cerebellar cortex is populated by inhibitory neurons such as Purkinje cells, basket cells, stellate cells, and Golgi cells, as well as excitatory neurons (granule cells and unipolar brush cells). Contrary to the recent interest in the TME-related functions of astrocytes, neurons in MB have not yet received much attention and only limited knowledge is available. It is, however, known that neurons can deliver mitogenic signals within the brain microenvironment to promote neuronal precursor cell growth [117]. You could speculate that this neuronal activity and release of paracrine factors stimulates the tumor growth of MB. This has already been proven in gliomas by the pioneering research of both the Michelle Monje lab and Frank Winkler lab. Venkatesh and colleagues of the Monje lab explained a mechanism of how neuronal activity promotes high-grade gliomal growth, showing that the upregulation of neuroligin-3 (NLGN3) in postsynaptic neurons promotes the proliferation of gliomal tumor cells through the induction of PI3K-mTOR signaling [118]. The shedding of NLGN3 from both oligodendrocyte precursor cells and neurons into the TME is mediated by the protease ADAM10 [119]. The use of ADAM10 inhibitors blocked the release of NLGN3 in the TME and inhibited gliomal growth in vivo, proving that neuronal-activity-regulated secretion is a targetable mechanism against tumor growth. Additional work from the Monje lab demonstrated that the integration of glioma cells into neural circuits via synaptic and electrical communication with neurons also promotes gliomal progression [120]. Electrochemical signaling within these circuits is mediated through potassium-evoked currents. This article was published back-to-back with Venkataramani et al. of the Winkler lab who showed that functional neuron-to-glioma synapses form a direct electrochemical communication between neurons and glioma cells [121]. Postsynaptic currents that are produced by these neuron-to-glioma synapses are mediated by glutamatergic AMPA receptors through which glioma invasion and growth are stimulated. Venkataramani et al. further advanced the understanding of the interactions between glioblastoma and neural circuits, demonstrating that glioblastoma cells can hijack neuronal mechanisms for brain invasion [122]. Distinct subpopulations of glioblastoma cells exist that resemble neuronal precursor cells, as demonstrated by scRNA-seq. These neuron-like glioblastoma cells drive tumor invasion and migration by adopting cellular mechanisms of neuronal development, such as branching migration, locomotion, and translocation. 

Lastly, several mutations have been associated with tumor-associated neuronal activity in gliomas. Germline *NF1* mutations in retinal neurons cause aberrant shedding of NLGN3, promoting the initiation of a low-grade glioma called optic pathway glioma [123]. Mutations in a PI3K-related gene, namely *PIK3CA*, have been revealed to selectively initiate brain hyperactivity in glioblastoma by secreting proteins such as the heparan sulfate proteoglycan GPC3 [124]. This protein is then able to trigger tumorigenesis by increasing the cellular proliferation through the acceleration of synapse formation and signaling. Both *NF1* and *PIK3CA* mutations are also found in a fraction of MB patients who mainly belong to the SHH-MB subgroup [125,126], which might suggest a potential relationship between neuronal-specific tumorigenic effects and MB. However, since both of these mutations are also often found in other non-CNS cancers such as colon, breast, and lung cancers [127,128], more research is needed into the link between these mutations and the tumorigenic effects of neurons in cancers such as glioma and MB. 

Overall, the studies described in this chapter could give some potential indications that neuronal-specific tumorigenic effects in MB may be underappreciated and could also have implications for MB formation and progression. Therefore, research on the tumorigenic activity of neurons in MB is needed.

## 6. Targeting the Brain Tumor Vasculature in Medulloblastoma

### 6.1. The Blood–Brain Barrier

Another unique feature of the brain TME is the presence of the BBB, a highly selective barrier between the systemic circulation and the brain. It comprises pericytes and astrocytic foot processes that surround specialized endothelial cells and microglia, which help regulate the BBB’s integrity [129]. The BBB protects the brain against circulating pathogens and toxic substances, although it also blocks the delivery of active pharmaceutical drugs. This proves a major challenge in the treatment of brain malignancies. 

In MB, the molecular subtype influences the BBB’s integrity and composition [130]. Phoenix et al. revealed that WNT-driven MB tumors, the subgroup with the best treatment response, induce an aberrant fenestrated vasculature through β-catenin paracrine signaling, making it possible for chemotherapeutic drugs to reach and accumulate in the tumor [130]. In contrast, SHH, group 3, and group 4 MBs have an intact BBB with no disruption of the endothelial tight junctions, which makes them less susceptible to chemotherapy. The authors demonstrated that treatment with Wnt7a restores a functional BBB in WNT-driven tumors, reducing the permeability of the chemotherapeutic drug vincristine. They indicated that chemotherapeutic drug delivery in MB could be improved if it is combined with agents that briefly open the BBB, such as nanoparticles [131]. Recently, multiple studies have proven the favorable effects of nanoparticle drug delivery for the treatment of both SHH-driven and group 3 MB [132,133,134,135,136]. Lipid–polymer hybrid nanoparticles, which are core–shell nanoparticles, were applied to prolong and increase the efficacy of a therapeutic smoothened targeting siRNA against SHH-MB [132]. The combination with microbubble-enhanced low-intensity focused ultrasound promoted the extravasation of the BBB and tumor. Furthermore, organic nanostructured materials such as liposomes [133] and polymeric micelles [134,135,136] have been utilized to improve the delivery of therapeutic agents over the BBB in SHH-MB tumors. Notably, polymeric micelles enhanced the therapeutic potential of vismodegib by reducing the bone toxicity and improving the central nervous system pharmacokinetics [135]. The latter was also used to study the effects of polymeric nanomedicine on group 3 MB cells [136]. However, only in vitro studies were performed with the group 3 MB cell line HD-MB03. 

Another method to briefly open the BBB is the use of cell-penetrating peptides. Synthetic HAV6 peptides caused transient, reversible fenestration in the BBB of group 3 MB tumor-bearing mice by targeting a specific His-Ala-Val (HAV) region on the extracellular domain of E-cadherin [137]. This region is essential for the formation of cadherin homodimer complexes that contribute to the physical restriction of the BBB. The simultaneous treatment of the HAV6 peptide with the BBB impermeable peroxiredoxin-1 inhibitor adenanthin reduced the tumor progression and increased the survival of mice bearing group 3 MB. 

Overall, the usage of drug delivery approaches such as nanoparticles, focused ultrasound, and cell-penetrating peptides will assist in bypassing the physical barrier that the BBB forms in non-WNT-MB subgroups, allowing for increased drug release at the tumor site within the CNS. This will be essential for increasing the efficacy of (TME-related) therapies in MB.

### 6.2. Angiogenesis in MB

Targeting brain tumor angiogenesis has also been proposed, since the primary tumors and metastases strongly depend on the aberrant vasculature’s organization [138]. However, only a few studies have investigated the vasculature in MB tumors. The anti-angiogenic compound axitinib, a VEGFR-1, -2, and -3 inhibitor, effectively reduced tumor growth and demonstrated a favorable toxicity profile in orthotopic group 3 MB models [139,140]. Additionally, the use of the anti-parasitic drug mebendazole has been proposed as a low-toxicity treatment for MB, as it inhibits angiogenesis via the inhibition of VEGFR-2 signaling [141]. Furthermore, it has been illustrated by Chan et al. that the treatment of MYC-associated MBs with the protein thrombospondin-1, which acts as an angiogenesis inhibitor, effectively led to reduced metastasis and increased survival in vivo [142]. However, clinical trials with anti-angiogenic treatments in MB patients have yet to be proven successful. The clinical evaluations of the anti-VEGF drugs bevacizumab [143,144,145] and PTC299 [146] have shown no significant improvement in the overall survival of MB patients. Therefore, follow-up studies are being conducted in patients with recurrent MB (Table 1; NCT01356290 and NCT04743661). Additionally, clinical trials are currently being performed targeting other angiogenic-associated targets, such as VEGFR-2 (Table 1; NCT04501718), FGFR (Table 1; NCT03155620), and COX2 (Table 1; NCT03257631 and NCT01661400), in patients with recurrent MB and other pediatric brain malignancies.

### 6.3. Cerebrospinal Fluid Circulation and the Lymphatic System

Besides the BBB and angiogenesis, the involvement of the cerebrospinal fluid (CSF) circulation in the TME of MB cannot be overlooked. Leptomeningeal dissemination (LMD) through the CSF is the main route of MB metastasis and the primary cause of mortality in MB patients [147]. LMD has a higher incidence in group 3 and group 4 MB patients and is observed in 10–30% of SHH-MB patients, whereas patients with WNT-driven MB rarely show metastases [3,148]. Several biological processes of LMD in MB have been discovered, with both intra- and intercellular signaling mechanisms. Molecular mechanisms of LMD have been reviewed by Li and colleagues [149]. Here, we will only discuss the intercellular processes associated with TME interactions. It has been shown that the abnormal expression of the transcription factor ATOH1 can promote LMD in SHH-driven MB through the aberrant activation of genes involved in cytoskeleton and extracellular matrix remodeling, such as *PDGFB* and *PDGFRB* (Figure 3A) [150]. ATOH1 plays a key role in SHH-MB by mediating the transit proliferation of granule cell precursors and suppressing differentiation in response to SHH, which is secreted by Purkinje cells, MB cells, and tumor-associated astrocytes [151,152]. Upon activation, ATOH1 can control the formation of primary cilia by transcriptionally regulating Cep131, which allows for SHH-triggered proliferation [152]. Cep131 is a protein responsible for facilitating the integrity of centriolar satellites, small non-membranous cytoplasmic granules that localize and move around the centrosomes and cilia. Furthermore, it has been found that SHH prevents the degradation of ATOH1 via the E3 ubiquitin ligase HUWE1, resulting in a positive autoregulatory feedback loop between SHH and ATOH1 (Figure 3A) [153]. Consequently, metastatic tumors arising through this process contain a strong dependency on the SHH signaling pathway [150]. Furthermore, Martirosian and colleagues explained that metastatic MB cells use the enzyme GABA transaminase (ABAT) to survive in the nutrient-deficient CSF microenvironment by metabolizing GABA, an inhibitory neurotransmitter, as an energy source (Figure 3B) [154]. Through this mechanism, metastatic MB cells exploit differentiated GABAergic neuronal characteristics such as GABA metabolism, histone deacetylation, and decreased tumor cell proliferation, thereby promoting LMD. 

In a recent study, it was also observed that primary and metastatic patient-derived MB tumors a contain reduced expression of the GABA_A_ receptor (Figure 3B) [155]. Decreased GABA_A_ receptor activity results in fewer interactions between GABA molecules with their receptors, causing more circulation of free GABA in the body, which can be exploited for neurotransmitter-mediated tumor progression and ABAT-mediated metabolism. Additionally, also a hematogenous route for MB LMD was discovered (Figure 3C) [156]. Metastatic circulating MB cells were found in the blood of therapy-naïve patients. Flank xenografting and parabiosis mice studies were performed to confirm that circulating MB cells can spread through the blood to the leptomeninges. An analysis of leptomeningeal metastases that originated through hematogenous dissemination and CSF samples collected from patients with group 3 and group 4 MB both detected high levels of CCL2, which is needed to drive LMD via the activation of the CCL2–CCR2 axis [156,157]. In addition, higher levels of CXCL1, IL-6, and IL-8 were observed in MYC-amplified group 3 MB patients [157]. More in-depth evaluations of the CSF of recurrent MB patients revealed that proteins (GPR37, ADAMTS1, and GAP43), metabolites, and lipids (tryptophan, methionine, serine, triglycerides) indicative of tumor-associated hypoxia were upregulated compared to the CSF of patients without cancer [158,159]. These studies point out that analysis of the CSF can reveal markers of metastatic MB progression and give insights into the impact of MB on the CSF microenvironment. A method to specifically target leptomeningeal metastases was studied by Engelhard et al., who used etoposide-bound magnetic nanoparticles (Etop-MNPs) to remotely target tumor cells [160]. They performed studies by mimicking the CSF drug delivery of Etop-MNPs in vitro and examined their cytotoxic effects on D283 cells, a human metastatic MB cell line. The novel Etop-MNPs killed the D283 cells in a dose-dependent manner, illustrating the potential for this novel treatment option. However, further studies in animal models are needed to investigate the effects of this treatment modality against MB, which was also noted by Engelhard et al.

Lastly, the role of the lymphatic system in the brain and its involvement in brain tumors has been gaining more attention [161]. It has recently been found that a lymphatic network surrounds the brain, where it is involved in the clearance of waste and macromolecules and the drainage of cerebral spinal fluid [19,20,162,163]. However, tumor cells also use lymphatic vessels to migrate and metastasize, which has implications for treatments. Interestingly, the main lymphatic endothelial cell growth factor VEGFC, which promotes the sprouting of lymphatic vessels, has been associated with anti-tumoral effects in MB [164]. This indicates that the lymphatic system might be beneficial in preventing MB growth as opposed to the current understanding in other cancers. 

## 7. Conclusions

The TME plays a vital role in the initiation and progression of MB. An intricate multicellular network of tumor and progenitor cells, immune cells, astrocytes, neurons, supporting stromal cells, blood and lymphatic vessels, and ECM components all impact tumor growth in their own way. Although there is great interest in studying the effects of the TME on cancer development, a clear representation of the complex TME interactions in MB has yet to be made. One of the reasons that complicates these studies is the age effect of MB, being a pediatric tumor. Therefore, it is essential to expand our current knowledge of the young TME related to adult TME. In addition, it will be crucial to further investigate the individual brain TME components in more complex microenvironments to study the cellular and non-cellular interactions in MB, especially since the brain TME contains unique brain-resident TME components, such as astrocytes, neurons, and microglia, which thrive on cell–cell interactions. Furthermore, it will be important to increase our understanding of how all parts of the TME in MB are affected by both standard-of-care treatment and novel therapeutic options. The development of improved in vitro models such as co-culture systems, organoids, and other multi-cellular model systems will be crucial to assess these (novel) drug targets. A better understanding of the complex TME in MB will also be needed to fully exploit the full potential of novel treatment approaches such as immunotherapy. The critical steps to be considered in assessing the clinical safety and efficacy of novel immunotherapy approaches consist of accurate patient stratification and the identification of appropriate predictive biomarkers and candidate targets. One of the main challenges for the treatment of MB remains the presence of the BBB, which complicates drug delivery into the brain. Novel drug delivery approaches such as the use of nanoparticles, focused ultrasound, and cell-penetrating peptides, will assist in bypassing the physical barrier of the BBB and will eventually be essential for increasing the efficacy of (TME-related) therapies in MB. When we are able to get brain-targeted therapies past this barrier, many interesting TME components will be available that form putative targets, such as the ECM components, microglia, astrocytes, and neurons. This shows that there is great therapeutic potential for TME-related treatments of MB. 

## Figures and Tables

**Figure 1 cancers-14-05009-f001:**
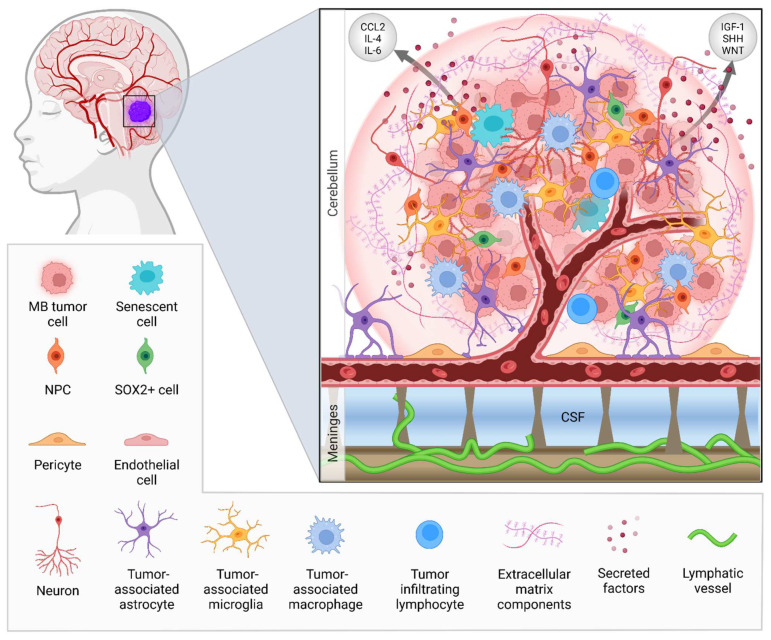
The tumor microenvironment of human medulloblastoma (MB). A MB tumor consists of an intricate multicellular network with extracellular matrix (ECM) components. These tumors are thought to arise from embryonic progenitor cells such as neural progenitor cells (NPCs). Senescent cells and SOX2+ cells form modes through which recurrence or relapse can occur. The blood–brain barrier (BBB) is an important part of the tumor microenvironment (TME) of MB. It consists of pericytes and astrocyte foot processes that surround the blood vessels, which are composed of a basement membrane and specialized endothelial cells. In most MBs the BBB is intact, functioning as a highly selective border. However, this integrity is influenced by the MB subgroup. The defining pattern of metastasis for MB is leptomeningeal dissemination, in which tumor cells can spread to the meninges through the cerebrospinal fluid (CSF). The lymphatic vasculature in the brain has only been recently discovered [19,20]. The role of lymphatic vessels in MB has, therefore, not been studied yet. However, this system might prove to be an important asset of the TME of MB. Furthermore, the TME of MB consists of unique brain-resident cell types such as microglia [21], astrocytes [22], and neurons [12], each associated with MB tumor progression in distinctive ways. MB has a low occurrence of tumor-infiltrating lymphocytes (T and B cells) and other immune cells [23]. Most abundant are the tissue-resident microglia and tumor-associated macrophages. All cells within the TME are responsible for the production of secreted factors such as growth factors, cytokines, and chemokines (e.g., IL-4, IL-6, CCL2, IGF-1, SHH, WNT) that stimulate tumor progression [22,24,25]. Support for this multicellular tumor network is provided by ECM components. MBs are highly enriched in glycoproteins (e.g., laminin and vitronectin) and proteoglycan-degrading enzymes (e.g., heparanase), causing dysregulation of the brain’s ECM composition.

**Figure 2 cancers-14-05009-f002:**
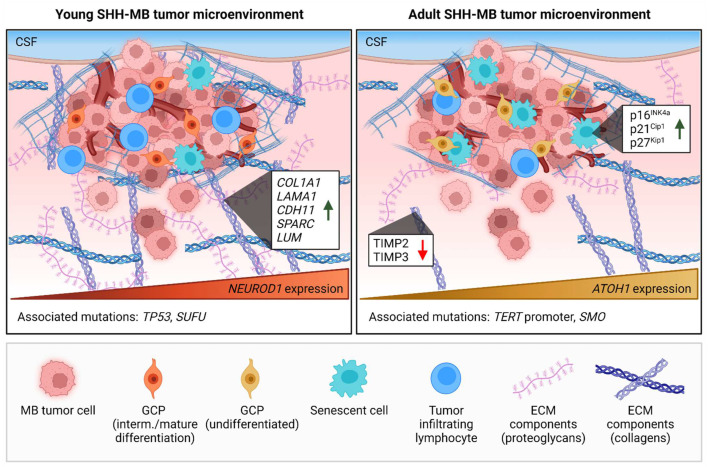
Age-associated differences in the tumor microenvironments of young and adult human SHH-driven medulloblastoma (MB). Age affects both the cellular and extracellular matrix (ECM) compositions of the MB tumor microenvironment (TME). Pediatric SHH-MB tumors contain relatively more tumor-infiltrating lymphocytes [23], whereas dormant senescent cells are more associated with the development of SHH-MB tumors in adults [33,44]. Preneoplastic MB lesions in adults show increased expression of senescence markers p16^INK4a^, p21^Cip1^, and p27^Kip1^. Additionally, young and adult SHH-MB tumors display varying orders of granule cell precursor (GCP) differentiation. *NEUROD1^+^* GCPs in pediatric SHH-MB is a sign of an intermediate to mature differentiation state, while *ATOH1* expression marks an undifferentiated state of GCPs in adult SHH-MB. Furthermore, ECM components, such as proteoglycans and collagens, are enriched in pediatric SHH-MB tumors. This has been observed both by an increase in ECM-related genes (e.g., *COL1A1*, *LAMA1*, *CDH11*, *SPARC*, *LUM*) in pediatric SHH-MB, as well as a decrease in ECM enzymatic inhibitors (TIMP2 and TIMP3) in adult SHH-MB. Lastly, *TP53* and *SUFU* mutations are more associated with pediatric SHH-MB, whereas *TERT* promoter and *SMO* mutations are mainly found in adult SHH-MB tumors. CSF = cerebrospinal fluid. Green arrow = increased expression; red arrow = decreased expression.

**Figure 3 cancers-14-05009-f003:**
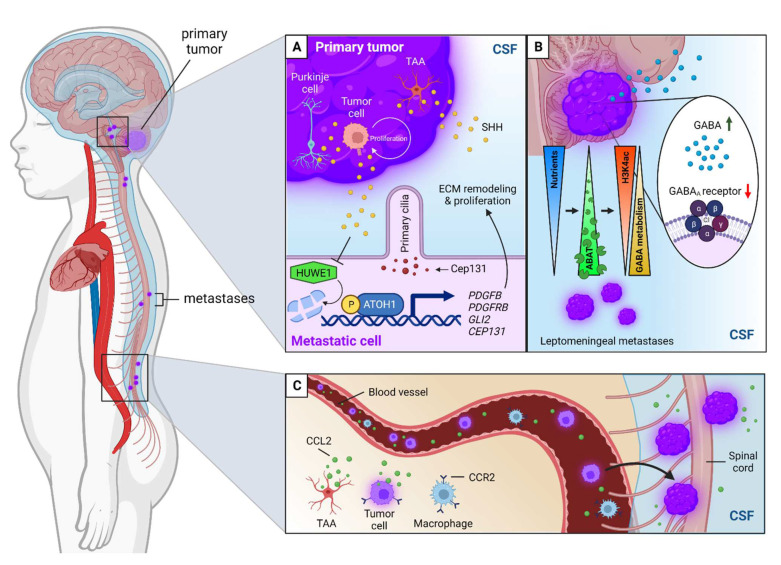
TME-associated mechanisms of leptomeningeal dissemination (LMD) in medulloblastoma (MB). (**A**) Aberrant SHH signaling and ATOH1 expression promote LMD in SHH-MB. SHH ligand is produced by Purkinje cells, MB tumor cells, and tumor-associated astrocytes (TAA) and stimulates the continuous proliferation of tumor cells through the activation of GLI and ATOH1 transcription factors. ATOH1 can control the formation of primary cilia by regulating the expression of Cep131, allowing for SHH-triggered proliferation, and also influences extracellular matrix (ECM) remodeling by activating ECM remodeling genes such as *PDGFB* and *PDGFRB* enabling metastatic spread. Additionally, SHH prevents the degradation of ATOH1 by blocking HUWE1. (**B**) Metastatic MB cells use GABA transaminase (ABAT) to metabolize GABA to survive in the nutrient-deficient cerebrospinal fluid (CSF). The metastatic cells adapt mature GABAergic neuronal characteristics such as H3K4ac histone deacetylation and GABA metabolism. Furthermore, GABA_A_ receptor activity is decreased in the primary MB tumor and leptomeningeal metastases, yielding more free circulating GABA molecules in the body. Green arrow = increased availability; red arrow = decreased activity. (**C**) LMD can also occur via a hematogenous route via the activation of the CCL2–CCR2 axis. CCL2 is, among others, secreted by TAAs and tumor cells. Both tumor cells and (tumor-associated) macrophages contain CCR2 receptors for CCL2 signaling. The CCL2 levels are increased in the CSF of MB patients.

**Table 1 cancers-14-05009-t001:** Currently ongoing clinical trials targeting TME components of medulloblastoma.

NCT	Phase	Treatment Modality	Drug	Target	Patient Enrollment	Age Group	Primary Outcome
NCT02359565	1	ICI	Pembrolizumab	PD-1	MB, EP, HGG, DIPG, HypBT	Pediatric, AYA	Safety, ORR, PD-1+ T-cell change
NCT02813135	1–2	ICI	Nivolumab lirilumab	PD-1 and KIR2DL1/ KIR2L3	MB, EP, HGG, DIPG	Pediatric	ORR, TTP
NCT03173950	2	ICI	Nivolumab	PD-1	MB, EP, CPC, A/MM, PRT	Adult	6-month PFS, ORR
NCT02793466	1	ICI	Durvalumab	PD-L1	MB, EP, HGG, DIPG	Pediatric, AYA	Safety, MTD
NCT04049669	2	ICI	Indoximod with radiation and chemotherapy	IDO	MB, EP, HGG, DIPG	Pediatric	8-month PFS, 12-month OS
NCT05106296	1	ICI and SMI	Indoximod with ibrutinib and chemotherapy	IDO and BTK (respectively)	MB, EP, HGG, PNET	Pediatric	ORR, toxicity
NCT03389802	1	ICI	Agonistic monoclonal antibody APX005M	CD40	MB, EP, HGG, DIPG	Pediatric	Safety, RP2D
NCT03500991	1	ACT	Anti-HER2 CAR T-cells	HER2^+^	MB, EP, HGG, CPC, PNET, ATRT	Pediatric, AYA	Safety and feasibility
NCT03638167	1	ACT	Anti-EGFR806 CAR T-cells	EGFR^+^	MB, EP, HGG, CPC, PNET, ATRT	Pediatric, AYA	Safety and feasibility
NCT03652545	1	ACT	Tumor multi-antigen associated T-cells	PRAME^+^, WT1^+^ and/or BIRC5^+^	MB, EP, HGG, CPC, DIPG	All	Safety and feasibility, MTD
NCT04099797	1	ACT	Anti-GD2 CAR T-cells	GD2^+^	MB, EP, HGG, DIPG	Pediatric	MTD
NCT05298995	1	ACT	Anti-GD2 CAR T-cells	GD2^+^	MB, HGG, DIPG	Pediatric, AYA	Safety, MTD
NCT04185038	1	ACT	Anti-B7-H3 CAR T-cells	B7-H3^+^	MB, EP, DIPG, PB, CPC, PNET, ATRT	Pediatric, AYA	Safety and feasibility
NCT04510051	1	ACT	Anti-IL13Rα2 CAR T-cells	IL13Rα2^+^	Recurrent/refractory brain tumors	Pediatric, AYA	Safety and feasibility
NCT04661384	1	ACT	Anti-IL13Rα2 CAR T-cells	IL13Rα2^+^	MB, EP, HGG	Adult	Safety and feasibility, 3-month OS
NCT05131763	1	ACT	Anti-NKG2D CAR T-cells	NKG2DL^+^	MB, HGG	Adult	Safety/toxicity
NCT01326104	2	ACT	Chemotherapy followed by treatment with total tumor RNA-loaded dendritic cells	Patient- specific antigens	MB, PNET	Pediatric, AYA	12-month PFS
NCT03299309	1	Vaccine	PEP-CMV vaccine	CMV pp65	MB, HGG	Pediatric, AYA	Safety
NCT05096481	2	Vaccine	PEP-CMV vaccine	CMV pp65	MB, HGG, DIPG	Pediatric	4-month PFS
NCT04978727	1	Vaccine	SurVaxM vaccine	Survivin (BIRC5)	MB, EP, HGG, DIPG, AA, AOD	Pediatric	Safety/toxicity
NCT02962167	1	Oncolytic virus	Modified measles virus (MV-NIS)	Na^+^/I^−^ symporter	MB, ATRT	Pediatric, AYA	Safety, RP2D
NCT03911388	1	Oncolytic virus	Engineered HSV G207	Cytopathic effect	MB, EP, HGG, PNET	Pediatric	Safety
NCT03043391	1	Oncolytic virus	Recombinant polio/rhinovirus (PVSRIPO)	CD155	MB, EP, HGG, ATRT, AA, AOA, AOD	AYA	Safety/toxicity
NCT02444546	1	Oncolytic virus	Engineered wild-type reovirus with GM-CSF	Cytopathic effect	MB, HGG, DIPG, AA, AOD, ATRT, PNET	AYA	MTD
NCT04758533	1–2	Oncolytic virus	Optimized adenovirus (ICOVIR-5)	pRB pathway	MB, DIPG	Pediatric	Safety, efficacy, MTD
NCT01356290	2	MAB	Bevacizumab with chemotherapy	VEGF	MB, EP, ATRT	Pediatric	Efficacy
NCT04743661	2	MAB	Bevacizumab with omburtamab and chemotherapy	VEGFB7-H3^+^	MB, EP	Pediatric	2-year EFS
NCT04501718	2	SMI	Apatinib with chemotherapy	VEGFR-2	Recurrent MB	Pediatric	ORR, PFS, OS
NCT03155620	2	SMI	Erdafitinib	FGFR	Recurrent/refractory pediatric tumors with mutations	Pediatric	ORR
NCT03257631	2	IMiD	Pomalidomide	COX-2 and cereblon	MB, EP, HGG, DIPG	Pediatric, AYA	ORR or long-term SD
NCT01661400	1	IMiD	Thalidomide	COX-2 and cereblon	MB, EP, HGG, DIPG	Pediatric, AYA post-transplant	Safety, stem-cell transplant-related toxicity

Obtained from https://clinicaltrials.gov/; current as of 24 July 2022. AA = anaplastic astrocytoma; ACT = adoptive cellular therapy; A/MM = atypical/malignant meningioma; AOA = anaplastic oligoastrocytoma; AOD = anaplastic oligodendroglioma; ATRT = atypical teratoid rhabdoid tumor; AYA = adolescent and young adult; CPC = choroid plexus carcinoma; DIPG = diffuse intrinsic pontine glioma; EFS = event-free survival; EP = ependymoma; HGG = high-grade glioma; HypBT = hypermutated brain tumor; ICI = immune checkpoint inhibitor; IMiD = immunomodulatory drugs; MAB = monoclonal antibody; MB = medulloblastoma; MTD = maximum tolerated dose; NTC = national clinical trial; ORR = objective response rate; OS = overall survival; PB = pineoblastoma; PFS = progression-free survival rate; PNET = primitive neuroectodermal tumor; PRT = pineal region tumor; RP2D = recommended phase 2 dose; SD = stable disease; SMI = small molecule inhibitor; TTP = time to progression.

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
