# Peer review of "The Tumor Microenvironment of Medulloblastoma: An Intricate Multicellular Network with Therapeutic Potential"

_cancers, 2022, doi:10.3390/cancers14205009_

Round 1

Reviewer 1 Report

The review proposed by Dr. van Bree Wilhelm represents a significant  contribution to the Medulloblastoma field, since it propose to comprehensively analyze the different aspects of the tumor micoenvironment, rather than focusing on the single aspects. Considering the growing importance of this topic in cancer research and the strong cross-talk between the different cellular components, their effort provide a useful up to date guide for researchers working on specific aspects of medulloblastoma. The reviews reads very well, the images are appropriate and of very good qulity. I only have a couple of minor suggestions:

-Line 50: it would be probably helpful for the reader a brief highlight even on the other precursors, whose transformation leads to non-SHH MB, besides discussing the origin of SHH type (see for instance Williamson et al., 2022, Cell Reports 40, 111162)

-Line 138: to my taste, this statement sound a little too 'finalistic'.  I would rephrase it like this:

The development of senescence in malignant cells, avoiding growth and detection etc.

- Line 141: I would have probably expected 'mechanism of progression' instead of 'hypothesis of recurrence'

Reviewer 2 Report

In the review article-The tumor microenvironment of medulloblastoma: an intricate multicellular network with therapeutic potential-authors discussed the components of the Medulloblastoma (MB) tumour microenvironment and pointed (in abstract, and only in  few sections of the manuscript) that the age effect of TME (young versus adult MB) is essential to expand the current knowledge. Indeed, the authors have made great efforts to pile up the information from the literature, however, I believe some critical information’s are missing here:

  • While the authors want to highlight TME in MB, they remain focused on TME more generally, without providing any perspective on what exactly to expect in young/adult TME in MD or CNS. And why only age, A wide range of (genetic) subgroups with presumably different TME components continue to pose a challenge for MB-associated therapies (especially immunotherapies).
  • MBs are considered as "cold" due to the differential expression of cancer-specific antigens on their cell surface, at least two important are- PRAME, which can triggers autologous T cell–mediated immune responses and is known to correlate well with the overall survival groups. And the second is cancer testis antigens (CTAs), the best-described antigens in MBs. I have not found any information about them in the article.
  • In addressing such a critical issue as TME in MD, the authors should have taken the liberty to extend their approach to include CNS/Brain TME (because of its intrinsic immunosuppressive properties). For instance, oncogenic transcription factors of the MYC family (c-MYC, MYCL, and MYCN) are frequently deregulated in MBs and are equally important for the CNS development. Since MYCs offer therapeutic opportunities (via epigenetic drugs/PRMT5 inhibitors), the authors should have collected some information to provide a viewpoint on the possibility concerning connecting these approached in young and/or adult MBs.
  • Section 4.1, contains the table from clinical trials in MB and bit descriptive information into the text. The readers would expect some conclusions to be drawn, because to me all the included studies looked like safe/good PFS-OS with convincing results and worth pursuing further.
  • In section 6.1, the authors discussed the role BBB in MB therapy. I do understand that several technologies including nano particles are being tested for BBB, however, in this particular section, I don't understand the connection between these methods in MB and TME. If the authors are suggesting that drug release at a specific site in the CNS is a prerequisite to tackle the complexity of TME in different types of MB, please elaborate.
  • Can you please provide the references for lines 111-116 (in Figure 1), 187-188 (Figure 2), line 224-226? In my opinion, when compiling such compact figures (like Figure 1, 2), it is very important to indicate the correct sources (human or mouse medulloblastomas etc.).
  • Please recheck line 470-471, I think besides SHH, there are also some updates about astrocytes in WNT/group 3 and group 4 MBs. Also, please elaborate your statement - a fraction of SHH-MB 462 cells into tumor-associated astrocytes is a novel mechanism through which tumor progression and relapse are promoted (Line 462-463). Do you mean SHH-MB associated mutations + tumor-associated astrocytes for suppressing the tumor progression? It is important to mention that mutations are usually overlapping in cancers, please see my next comment in this context.
  • Line 512- , NF1 and PIK3CA mutations are also present in lung cancer, they are secondary mutations, so it is not surprising that they are present in some of the SHH-MB subgroupsIn my opinion, the hypothesis of neuronal-specific tumorigenic effects in MB is not completely wrong, but should be discussed (rather than claimed) in the context of an ongoing open discussion.
  • Line 479-480, it this authors personal opinion?
  • The conclusion is very unimpressive. At this stage, the article looks like a collection of past developments, but offers no insights into further research in the field of MBs.

Round 2

Reviewer 2 Report

To my opinion the revised manuscript can be accepted for publiucation in the current form.

Author Response

To my opinion the revised manuscript can be accepted for publiucation in the current form.

Response: We would like to sincerely thank the reviewer for giving their approval for publication of our manuscript after critically reviewing our manuscript and providing valuable suggestions and comments.